# Association between Mortality and Lung Low Attenuation Areas in NSCLC Treated by Surgery

**DOI:** 10.3390/life13061377

**Published:** 2023-06-12

**Authors:** Davide Colombi, Ganiyat Adenike Ralitsa Adebanjo, Rocco Delfanti, Sara Chiesa, Nicola Morelli, Patrizio Capelli, Cosimo Franco, Emanuele Michieletti

**Affiliations:** 1Department of Radiological Functions, Radiology Unit, AUSL Piacenza, Via Taverna 49, 29121 Piacenza, Italy; nicola.morelli.md@gmail.com (N.M.); e.michieletti@ausl.pc.it (E.M.); 2Department of Medicine and Surgery (DiMeC), Scienze Radiologiche, University of Parma, Via Gramsci 14, 43126 Parma, Italy; g.adenikeadebanjo@gmail.com; 3Department of Surgery, General Surgery Unit, AUSL Piacenza, Via Taverna 49, 29121 Piacenza, Italy; r.delfanti@ausl.pc.it (R.D.); p.capelli@ausl.pc.it (P.C.); 4Emergency Department, Pulmonology Unit, AUSL Piacenza, Via Taverna 49, 29121 Piacenza, Italy; s.chiesa3@ausl.pc.it (S.C.); c.franco@ausl.pc.it (C.F.)

**Keywords:** nonsmall-cell lung cancer, pulmonary emphysema, survival

## Abstract

Background: to test the association with overall survival (OS) of low attenuation areas (LAAs) quantified by staging computed tomography (CT) of patients who underwent radical surgery for nonsmall-cell lung cancer (NSCLC). Methods: patients who underwent radical surgery for NSCLC at our institution between 1 January 2017 and 30 November 2021 were retrospectively evaluated. Patients who performed staging or follow-up CTs in other institutions, who received lung radiotherapy or chemotherapy, and who underwent previous lung surgery were excluded. At staging and 12-months follow-up CT, LAAs defined as voxels <−950 Hounsfield units, were extracted by software. The percent of LAAs relative to whole-lung volume (%LAAs) and the ratio between LAAs in the lobe to resect and whole-lung LAAs (%LAAs lobe ratio) were calculated. Cox proportional hazards regression analysis was used to test the association between OS and LAAs. Results: the final sample included 75 patients (median age 70 years, IQR 63–75 years; females 29/75, 39%). It identified a significant association with OS for pathological stage III (HR, 6.50; 95%CI, 1.11–37.92; *p* = 0.038), staging CT %LAAs ≥ 5% (HR, 7.27; 95%CI, 1.60–32.96; *p* = 0.010), and staging CT %LAA lobe ratio > 10% (HR, 0.24; 95%CI 0.05–0.94; *p* = 0.046). Conclusions: in patients with NSCLC who underwent radical surgery, a %LAAs ≥ 5% and a %LAA lobe ratio > 10% at staging CT are predictors, respectively, of shorter and longer OS. The LAA ratio to the whole lung at staging CT could be a critical factor to predict the overall survival of the NSCLC patients treated by surgery.

## 1. Introduction

Surgery is the treatment of choice for nonsmall-cell lung cancer (NSCLC) with stage I and II and for patients with a noncentrally located resectable tumor in the absence of nodal metastasis at both computed tomography (CT) and positron emission tomography (PET) [1,2]. The survival of patients with NSCLC treated surgically is determined by various parameters, including age, stage, tumor size, tumor extent, lymph node stage, and positive lymph node number [3,4]. In patients affected by advanced squamous cell carcinoma higher visual score emphysema is associated with a twofold risk increase of death [5]. The improvement of lung function after resection of cancer in patients with emphysema has been widely demonstrated [6,7]. When the tumor is located in highly emphysematous areas, cancer resection combined with “lung volume reduction surgery” (LVRS) determines an increase in the postoperative lung function [6]. Furthermore, in patients with emphysema who underwent lobar resection for cancer without LVRS, postoperative pulmonary function is better than predicted in previous surgery [7]. In particular, patients with more pure obstructive airway disease with low forced expiratory volume in 1 s (FEV1) and low FEV1/forced vital capacity (FVC) ratio, have a higher likelihood to benefit in pulmonary function after lobectomy [8]. The presence of emphysema is a predictor of death for lung cancer [9,10]. Nevertheless, the emphysema quantification at CT was estimated visually with semiquantitative methods or with the software aid with a binary approach (presence or absence of emphysema) [9,10]. The impact of software-aided emphysema quantification at CT (QCT) on overall survival (OS) at the staging CT of patients resected for NSCLC has not been fully investigated [11]. The percentage relative to the whole lung of low attenuation areas (<−950 Hounsfield units, HU; LAAs) evaluated by quantitative CT are related to death in the smoking population [12,13]. In addition, lobar quantification of LAAs is one of the required steps for planning advanced emphysema treatment with the bronchoscopic placement of endobronchial valves [14]. Furthermore, the lobar quantification of LAA may be the method of choice to predict lung function after surgery in patients affected by NSCLC [15]. Thus, the aim of our study was to test the association with OS of LAAs quantified at staging CT in patients who underwent radical surgery for NSCLC. In particular, we aimed to test the prognostic significance in terms of OS for both LAAs extent relative to whole-lung volume (considering three groups of patients on the basis of LAA extent: %LAAs ≤ 0.5%, %LAAs 0.5–5%, and %LAAs ≥ 5%) and the LAAs volume of the lobe to resect relative to whole-lung LAAs volume.

## 2. Materials and Methods

### 2.1. Study Population

This retrospective study was approved by the Local Ethics Committee (institutional review board -IRB- approval number 897/2022/OSS*/AUSLPC). Informed consent was obtained from all subjects involved in the study; as stated by the IRB, informed consent was waived when the patient could not be reached in a timely manner owing to organizational reasons. This study included 189 patients (median age 70 years old, interquartile range, IQR: 63–76 years; males 124/189, 66%) who underwent radical surgery for NSCLC in our hospital between 1 January 2017 and 30 November 2021 [16]. The indication for radical surgical treatment was decided after a multidisciplinary board with the participation of a pulmonologist, thoracic surgeon, oncologist, radiotherapist, pathologist, and radiologist. All patients with clinical or radiological suspicion of lung cancer were discussed at the multidisciplinary board after the clinical-staging process, performed on the basis of the current guidelines of the European Society of Medical Oncology (ESMO) [1]. The classification of the T, N, and M categories was based on the eighth edition of the TNM classification for lung cancer [17]. The two main steps of the clinical staging were CT and PET-CT. In part-solid tumors, only the size of the solid part was used for the assignment of the T category. When additional nodules were present, the same lobe nodules were scored T3, ipsilateral but in a different lobe T4, and M1a in the case of contralateral nodules, considering them intrapulmonary metastases [18]. Nevertheless, when only two pulmonary foci were present, the conclusion of whether to consider a second primary or intrapulmonary metastases were obtained by the multidisciplinary tumor board [19]. Endosonography (endoscopic bronchial ultrasound, EBUS; endoscopic ultrasound, EUS) with needle aspiration was imperative when CT and PET-CT showed mediastinal nodal abnormality; nonetheless, even when CT and PET-CT were negative for mediastinal abnormality, EBUS\EUS was performed in case of ipsilateral hilar abnormality (cN1), of central tumors, and of peripheral tumors with ≥3 cm [1]. For the identification of brain metastases, we followed guidelines provided by the American College of Chest Physicians (ACCP) that suggest the use of brain magnetic resonance only in symptomatic patients and cancers in stage III/IV [20]. Surgery was proposed for all patients in stage I and II or noncentrally located tumors without nodal metastasis [1].

For surgical candidates, the risk of postoperative mortality and morbidity was based on the European Respiratory Society (ERS) and the European Society of Thoracic Surgery (ESTS) guidelines [21]. The preoperative evaluation of the FEV1 and diffusing capacity of the lung for carbon monoxide (DLCO) measurements were required. In the case of preoperative FEV1 or DLCO higher than 80% of the predicted value, pulmonary resection was not contraindicated. When FEV1 or DLCO were less than 80% of the predicted value, a more precise assessment with pulmonary exercise testing was performed. In the case of maximal oxygen consumption (VO2 max) higher than 20 mL/kg/min, the patient underwent surgical lung resection. When VO2 max was lower than 10 mL/kg/min, other treatment options were considered. If VO2 max was included between 10 and 20 mL/kg/min, it calculated the predicted postoperative both FEV1 (ppo-FEV1) and DLCO (ppo-DLCO). These values were calculated by the pulmonary segments to resect, taking into account the regional distribution of perfusion and ventilation. When ppo-FEV1 and ppo-DLCO were higher than 40% of the preoperative value, tumor resection was performed. 

Exclusion criteria were: 1. staging or follow-up CT 12 months after surgery (F-U CT) not performed in our hospital; 2. presence of significative CT artifacts; 3. thoracic radiotherapy (RT) or chemotherapy (ChT) performed previous surgery; and 4. previous lung surgery. Clinical data were obtained by medical records.

### 2.2. CT and LAAs Analysis

Both staging and F-U CT scans were performed with a 64-row CT scanner (Aquilon; Toshiba, Inc., Tokyo, Japan), and two different 16-row CT scanners (Emotion 16, Siemens AG, Forcheim, Germany; Brilliance 16, Philips Health Systems, Amsterdam, Netherlands). CT scans included the brain, chest, upper abdomen, and pelvis, obtained in the supine position. The chest was scanned with two phases: an unenhanced phase and a single-enhanced phase obtained after 35 s of peripheral intravenous power injection of 90 to 120 mL nonionic contrast material (Omnipaque 300 mg/mL iodine concentration; GE Healthcare, Little Chalfont, UK), based on patient size. LAAs quantification was performed on the unenhanced scan. The following parameters were used for the unenhanced scan: tube voltage range, 110 kV−130 kV; median tube current range, 75–378 mAs; pitch 1.5; and collimation, 1.25 mm. Sharp kernels were used to reconstruct imaging datasets with 1.5–2 mm slice thickness. 

Staging CT and F-U CT scans performed were anonymized and transferred to a dedicated workstation. A radiologist (D.C.) with eight years of experience quantified LAAs at staging CT and F-U CT by using a commercially available software (IntelliSpace Portal, version 12.1; Philips Health System). The COPD application automatically segmented the whole lung and the airways; later, after applying a noise reduction algorithm, were identified LAAs as the lung volume with voxels lower than −950 HU and calculated the percentage relative to whole-lung volume (%LAAs). Furthermore, the LAAs volume of each lobe (right upper lobe, middle lobe, right lower lobe, left upper lobe, and left lower lobe) was recorded [22]. Each patient was categorized on the basis of %LAA thresholds at staging CT (Figure 1) as provided by the statement of the Fleischner society: a. group 1 with trace emphysema when %LAAs was ≤0.5%; b. group 2 with mild emphysema when %LAAs was included within 0.5 and 5%; and group 3 with moderate emphysema when %LAAs was ≥5% [22]. The relative LAAs volume of the lobe that underwent resection as compared to the LAAs volume of the whole lung (%LAAs lobe ratio) was calculated with the following formula: (LAAs volume of the resected lobe/LAAs volume of the whole lung) × 100. A radiology resident (G.A.R.A.) with two years of experience quantified the %LAAs at F-U CT to evaluate the interreader agreement. The difference in %LAAs between staging CT and F-U CT was also calculated with the following formula: (%LAAs F-U CT − %LAAs staging CT)/%LAAs staging CT. In addition, an emphysema category upgrade between staging CT and F-U CT was recorded.

### 2.3. Lung Surgery

The American College of Chest Physicians (ACCP) guidelines were followed for the surgical treatment [23]. Surgery was indicated for patients without medical contraindications affected by NSCLC clinical stage I and II. Patients with NSCLC clinical stage I with comorbidities or borderline pulmonary function underwent sublobar resection instead of lobectomy. When sublobar resection was performed for lesions less than 2 cm, free margins larger than the maximum tumor diameter were achieved; in the case of lesions larger than 2 cm, free margins were higher than 2 cm. In patients with predominantly ground-glass lesions, shorter than 2 cm and in stage I, sublobar resection was performed instead of lobectomy. Postoperative platinum-based chemotherapy was performed in patients with a pathological stage higher or equal to stage II A,B (N1). Patients with positive bronchial margins after surgery underwent radiation therapy.

Mediastinal nodal dissection or sampling was performed in all patients, with the exception of patients affected by stage I NSCLC and intraoperative N0 status. Lung resection was performed by one thoracic surgeon (R.D.) with 17 years of experience in thoracic surgery. Patients underwent surgery by video-assisted thoracic surgery (VATS) through one or three port sites (1 cm to 3 cm). A chest tube of 20 F was positioned in the hemithorax before the closure of the chest wall, and it was removed one day after the air leaks disappeared. Following chest drainage removal, patients were discharged at home.

### 2.4. Statistical Analysis

Data are reported as median and IQR for continuous variables and absolute number with percentage for categorical variables. Patients were categorized into two groups according to death occurrence. Comparisons between variables were performed with the Mann–Whitney U test, Wilcoxon test, and Chi-square or Fisher’s exact test, as appropriate. The intraclass correlation coefficient (ICC) was calculated to assess inter-reader agreement. ICC was interpreted as follows: <0.40, poor agreement; 0.40–0.54, weak agreement; 0.55–0.69, moderate agreement; 0.70–0.84, good agreement; and 0.85–1.00, excellent agreement [24]. OS was calculated from the date of surgery to the time of death from any cause. Categories from continuous variables were obtained by using as thresholds the median value of the overall sample. Variables were correlated with OS using the Kaplan–Meier method (product limit). The survival functions were compared between independent groups of patients by means of the log-rank test. Univariable and multivariable Cox proportional hazards regression analysis was used to examine the association between prognostic variables and OS for estimating hazard ratios (HRs) and 95% confidence intervals (CIs). Variables with a *p* value < 0.1 at univariable analysis were included in the multivariable analysis using a backward regression model. A *p* value < 0.05 was considered significant. Statistical analysis was performed using MedCalc software, version 14.8.1 (MedCalc Software, Ostend, Belgium).

## 3. Results

### 3.1. Study Population

The final cohort included 75 patients (Figure 2) with a median age of 70 years old (IQR, 63–75 years), 29/75 (39%) were females. Demographics, comorbidities, and cancer details are summarized in Table 1. Other primary tumors were identified before or after NSCLC diagnosis in 26/75 (35%) patients. The majority of the patients were current or former smokers (61/75, 81%) with a median of 30 pack years (IQR, 11–40 pack years). Half of the patients were affected by COPD (31/75, 50%). Pathological stage IA was the most represented (53/75 patients, 71%); for the remaining patients 10/75 (13%) were in stage IB, 5/75 (7%) were in stage II, while 7/75 (9%) patients were in stage III. Adenocarcinoma was the most frequent cancer histology (60/75, 80%), followed by squamous-cell carcinoma (9/75, 12%).

Among demographics, comorbidities, and cancer characteristics, no significant differences were found between alive patients and nonsurvivors (Table 1). Nonsurvivors had a median age of 71 years old (IQR, 67–75 years old), similar (*p* = 0.230) to patients alive (69 years old, IQR, 62–75 years old). The majority of nonsurvivor patients were males (78% vs. 59%, *p* = 0.468), current or former smokers (89% vs. 80%, *p* = 1.000), in stage IA (67% vs. 71%, *p* = 0.716), and affected by adenocarcinoma (78% vs. 80%, *p* = 1.000). Furthermore, around one-third of patients who died had COPD (44% vs. 50%, *p* = 1.000) and a history of other cancer (44% vs. 33%, *p* = 0.719).

### 3.2. CT Analysis

LAAs quantification results are summarized in Table 1. At the staging CT, the median lung volume was 5.53 L (IQR, 4.79–6.39 L) and the median %LAAs was 0.12% (IQR, 0.03–0.82%). Considering the LAAs category, 52/75 (69%) patients showed trace emphysema, 14/75 (19%) mild emphysema, and 9/75 (12%) moderate emphysema. The median %LAAs lobe ratio was 10% (IQR, 0.25–24%). The median time elapsed between staging CT and F-U CT was 12 months (IQR, 11–13 months). Median F-U CT lung volume (4.70 L; IQR, 3.95–5.47 L) was lower when compared to staging CT (*p* < 0.001). Median %LAAs (0.30%; IQR, 0–2.02%) was similar between F-U CT and staging CT (*p* = 0.126). The inter-reader agreement was excellent for lung-volume quantification (ICC, 0.985; 95%CI, 0.976–0.991) while was good for %LAAs (ICC, 0.702; 95%CI, 0.492–0.796). The median difference in %LAAs between staging and F-U CT was −0.01% (IQR, −1.74–0.08%). Upgrade in the LAAs category at F-U CT in comparison to staging CT was observed in 19/75 (25%) of the patients.

Although not significant (*p* = 0.07), a trend towards a higher rate of moderate emphysema (LAAs ≥ 5%) was found in dead patients as compared to survivors (34% vs. 10%). At F-U CT, the rate of patients who upgrade the LAAs category was similar between the two groups of patients (22% for nonsurvivors vs. 26% for survivors, *p* = 1.000). No significant differences were demonstrated among patients who died and the remaining patients for the median %LAAs lobe ratio (9%, IQR 0.75–12 vs. 11%, IQR 0–25; *p* = 0.445) and for the difference in %LAAs between staging and follow-up CT scans (+0.11%, IQR −1.23 ± 15.12 vs. −0.04%, IQR −1.78 ± 0.10; *p* = 0.142).

### 3.3. Survival Analysis

The median survival time was 35 months (IQR, 26–49 months). During the observation time, 9/75 (12%) patients died. Kaplan–Meier curves showed significantly shorter OS for patients in pathological stage III (mean estimated OS 45 months vs. 63 months, *p* = 0.047; Figure 3a) and with moderate emphysema at staging CT (mean estimated OS 44 months vs. 64 months, *p* = 0.017; Figure 3b). Although not significant (*p* = 0.204), a trend towards longer survival was identified in patients with %LAAs lobe ratio at staging CT ≥ 10% (mean estimated OS 65 months vs. 58 months; Figure 3c).

Cox proportional hazards regression analysis results are shown in Table 2. At univariable analysis, pathological stage III (HR, 4.35; 95%CI, 1.02–21.06; *p* = 0.041), moderate emphysema at staging CT (HR, 5.27; 95%CI, 1.18–23.49; *p* = 0.029), and %LAA lobe ratio >10% (HR, 0.41; 95%CI 0.10–0.96; *p* = 0.048) were significantly associated with OS. By contrast, no significant association was found between OS and age >70 years old (HR, 1.75; 95%CI 0.47–6.52; *p* = 0.400), male gender (HR, 1.97; 95%CI 0.41–9.43; *p* = 0.397), history of other cancer (HR, 1.93; 95%CI 0.52–7.20; *p* = 0.326), current or former smoker condition (HR, 1.53; 95%CI 0.19–12.17; *p* = 0.687), pack years higher than 30 (HR, 1.24; 95%CI 0.31–5.01; *p* = 0.755), COPD status (HR, 0.85; 95%CI 0.23–3.17; *p* = 0.816), adenocarcinoma histology (HR, 1.51; 95%CI 0.30–7.41; *p* = 0.610), upgrade in LAAs category between staging and F-U CT (HR, 1.34; 95%CI 0.27–6.56; *p* = 0.712), and difference in %LAAs between staging and F-U CT higher than −0.01% (HR, 1.54; 95%CI 0.38–6.13; *p* = 0.541). Multivariable analysis confirmed a significant association with OS for pathological stage III (HR, 6.50; 95%CI, 1.11–37.92; *p* = 0.038), moderate emphysema at staging CT (HR, 7.27; 95%CI, 1.60–32.96; *p* = 0.010), and %LAA lobe ratio >10% (HR, 0.24; 95%CI 0.05–0.94; *p* = 0.046).

## 4. Discussion

In the present study, it was found a significant association between long-term mortality and LAAs quantified at staging in CT of patients affected by NSCLC who received surgery with curative intent. In particular, patients with %LAAs ≥5% of the whole-lung volume showed shorter overall survival; by contrast, patients with a ratio ≥10% between %LAAs of the resected lobe and whole-lung %LAAs showed longer overall survival.

A recent study performed on around 45,000 patients affected by NSCLC treated with surgery between 2004 and 2014, identified several risk factors for long-term (5 and 10 years after surgery) mortality [3]. Among demographics, increased age and male gender were associated with a higher risk of death [3]. Furthermore, several factors related to tumor burden or treatment, such as higher tumor size, mediastinal nodal involvement, or chemo-radiotherapy after surgery showed lower survival [3]. Accordingly, our study showed a significantly higher risk of death for patients in pathological stage III. Nevertheless, the association between mortality and LAAs was not investigated by Jia et al. [3].

Patients who undergo pulmonary resection for NSCLC affected by concomitant emphysema can benefit from the “LVRS effect” [25]. In patients with severe emphysema and lung cancer, tumor resection combined with LVRS provides significant benefit to lung function [6]. Furthermore, in the presence of emphysema, lung tumor resection performed without LVRS results in postoperative lung-function improvement [7]. The removal of poorly functioning and hyperexpanded lung tissue improves both chest wall and diaphragm mechanics, enhancing lung function.

During the last decades, the analysis of pulmonary lesions and lung parenchyma was the research subject for planning lung surgery in NSCLC. Several software can automatically segment pulmonary lesions, bronchial trees, and blood vessels; in addition, they can quantify mean lung density or calculate the emphysema index [26]. This computer-based assessment can affect lung surgery in terms of operability and resection strategy [26]. The estimation of postoperative lung function and the likelihood of mediastinal structures infiltration are better delineated with quantitative CT; for example, the lobar quantification of a well-aerated lung, defined as the lung volume included between −950 HU and −750 HU, at staging CT in patients affected by NSCLC, showed excellent and good agreement, respectively, with postoperative FEV1 and %DLCO measured 12 months after surgery [16,26]. Functional quantitative CT parameters can also affect the resection strategy whether to perform segmentectomy, bisegmentectomy, or lobectomy [26]. In addition, the quantification of %LAAs is useful for planning lung surgery in patients with NSCLC [15,27]. Patients with borderline lung function should need imaging-based quantification for predicting residual lung function after surgery [21]. Quantitative CT with the lobar analysis of normal functional lung with LAAs subtraction seems to be the method of choice to calculate predictive postoperative lung function [15,21]. 

Patients with higher %LAAs in the smoking population showed reduced OS [12,13]. Johannessen et al. identified %LAAs ≥10% as a strong predictor of all-cause long-term (8 years) mortality in a smoker population with 50% COPD prevalence [12].

Additionally, Lynch et al. demonstrated a significant association between long-term mortality (7 years) and %LAAs in a cohort of current or former smokers with at least 10 pack years of exposure to smoking [13]. Furthermore, in patients affected by advanced lung squamous-cell carcinoma not surgically treated, visual quantification of emphysema was correlated with shorter OS [5]. In patients affected by lung cancer, emphysema is a well-known risk factor for shorter OS [9,10]. Zulueta et al. demonstrated a higher risk of death for lung cancer in patients with marked emphysema, defined visually as more than one half of lung parenchyma [9]. In pathological stage I NSCLC, the presence of %LAA > 10%, identified patients with a twofold increased risk of death [10]. These findings are similar to the results obtained in the present study, which demonstrated reduced OS for patients with resected NSCLC and %LAAs ≥5%.

However, the literature data are lacking about the amount of %LAAs in the lobe to resect for NSCLC that provides advantages in terms of OS. For example, the lobar quantification of %LAAs at CT is a step required in the treatment of advanced emphysema [14]. Particularly, for the bronchoscopic placement of endobronchial valves, an extent of %LAAs > 30% in the target lobe is needed to successfully increase lung function; furthermore, the volume of the remaining lung should be assessed to guide treatment decisions [14]. Our study identified that a ratio ≥10% between %LAAs of the lobe to be resected relative to the whole-lung %LAAs is associated with longer OS in patients affected by NSCLC.

Our findings contribute to going beyond the binary presence or absence of emphysema for planning surgery in patients with lung cancer. Providing quantitative extent at CT of LAAs in both whole lung and in the lobe that should be resected, a more tailored approach for surgery could be achieved with more precise prognosis assessment and prediction of postoperative lung function, particularly for patients with borderline preoperative pulmonary-function tests. 

This study carries several limitations. First, it was a retrospective analysis from a single institution. Second, the measurements of lung volume and %LAAs can be affected by inspiratory levels, since adjustment for total lung volume (TLV) was not performed; this limitation could determine an overestimation of %LAAs, considering that a functional lung can be not completely ventilated due to suboptimal inspiration, with consequent reduced whole-lung volume [28]. Nevertheless, patients received detailed instructions for optimal inspiration before examinations and CT scans affected by inadequate inspiration were excluded [28]. Third, CT scans were performed with different scanners and kernels. LAAs quantification can be affected by several parameters among which are iterative reconstruction algorithm, software versions, and noise-reduction algorithm [29]. Particularly, the use of a noise-reduction algorithm reduces the quantification of %LAAs with better correlation to pulmonary function tests [29]. For this reason, we applied a noise-reduction algorithm in order to optimize emphysema quantification, despite CT being acquired with a different scanner and reconstructed with different kernels [29]. Fourth, the visual analysis of emphysema characteristics, as proposed by Lynch et al., was not assessed [22]. In smokers, visual scoring of thoracic CTs provides independent prognostic information for the clinical management of ever smokers; particularly mild centrilobular emphysema, moderate centrilobular emphysema, and confluent emphysema were associated with shorter OS. Nonetheless, the focus of our study was to assess the association of LAAs at quantitative CT with mortality in resected NSCLC, and not with qualitative emphysema analysis [22]. Fifth, the correlation with pulmonary-function tests was not investigated; however, the lung function of all patients was preserved at staging and permitted lung resection. Future prospective studies in a multicentric scale, including qualitative emphysema analysis and correlation with both preoperative and postoperative pulmonary function tests, are required to confirm and strengthen the present findings.

## 5. Conclusions

In patients affected by NSCLC suitable for surgery, a %LAAs ≥ 5% relative to whole lung at staging CT is associated with shorter overall survival; by contrast, longer overall survival has been demonstrated in case of a ratio ≥ 10% between %LAAs of the lobe to be resected and whole-lung %LAAs. The LAA ratio to the whole lung at staging CT could be a critical factor to predict the overall survival of the NSCLC patients treated by surgery.

## Figures and Tables

**Figure 1 life-13-01377-f001:**
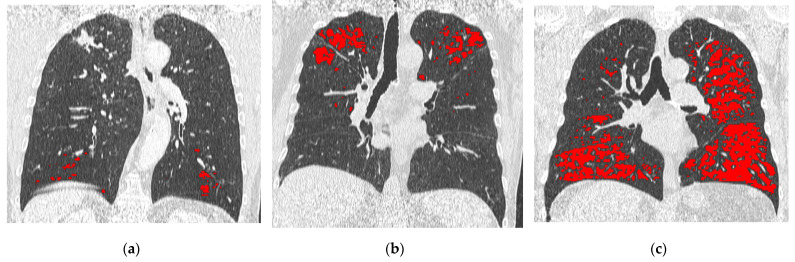
Example of patient categorization on the basis of %LAAs, highlighted in red at CT coronal multiplanar reconstruction: (**a**) Trace emphysema (%LAAs ≤ 0.5%); (**b**) Mild emphysema (%LAAs 0.5–5%); and (**c**) Moderate emphysema (%LAAs ≥ 5%).

**Figure 2 life-13-01377-f002:**
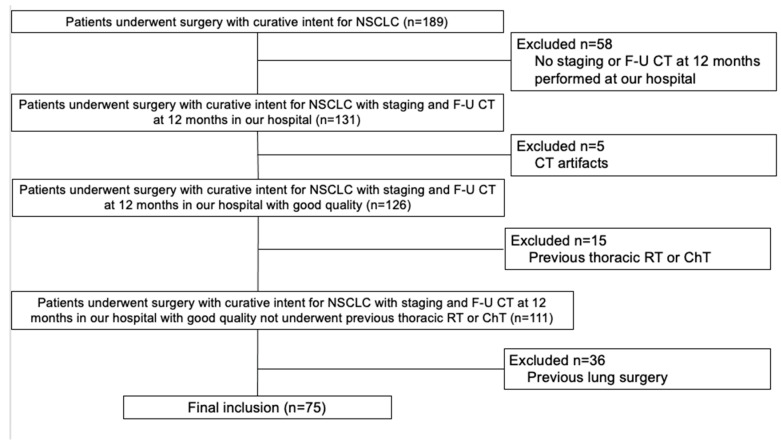
The diagram shows the patient selection process. Abbreviations: ChT, chemotherapy; CT, computed tomography; F-U, follow-up; NSCLC, nonsmall-cell lung cancer; RT, radiotherapy.

**Figure 3 life-13-01377-f003:**
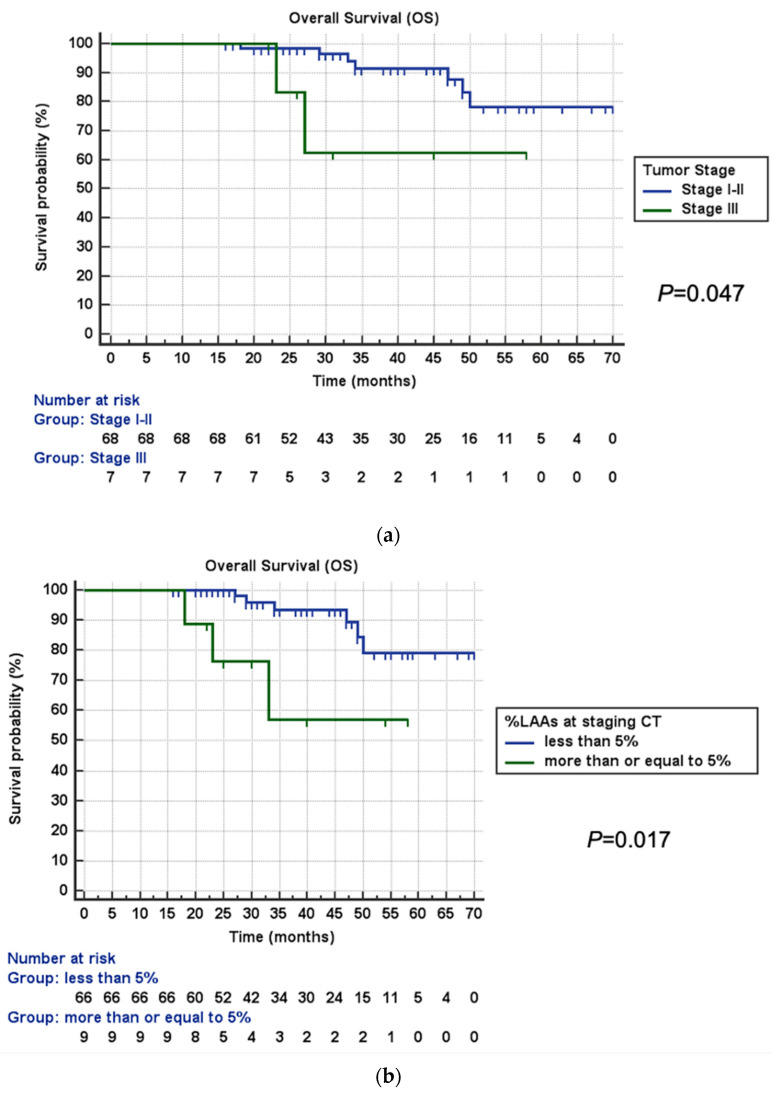
Kaplan–Meier estimates of overall survival (OS). (**a**) A significant association was identified between shorter OS and pathological tumor stage III (*p* = 0.047). (**b**) Patients with moderate emphysema (%LAA ≥ 5%) at staging CT showed significantly shorter OS (*p* = 0.017). (**c**) Although not significant (*p* = 0.204) was identified a trend towards longer OS for %LAAs lobe ratio ≥ 10% at staging CT. Abbreviations: CT, computed tomography; LAAs, low attenuation areas; OS, overall survival.

**Table 1 life-13-01377-t001:** Demographics, comorbidities, cancer characteristics, and LAA analysis according to patients’ survival.

Parameter	All Patients (*n* = 75)	Survivors (*n* = 66)	Non-Survivors (*n* = 9)	*p*-Value
Age (years)	70 (63–75)	69 (62–75)	71 (67–75)	0.230
Gender				
Male (*n*)	46/75 (61%)	39/66 (59%)	7/9 (78%)	0.468
Female (*n*)	29/75 (39%)	27/66 (41%)	2/9 (22%)	
History of other cancer than NSCLC (n)	26/75 (35%)	22/66 (33%)	4/9 (44%)	0.710
Smoking history				
Never (*n*)	14/75 (19%)	13/66 (20%)	1/9 (11%)	1.000
Current or former (*n*)	61/75 (81%)	53/66 (80%)	8/9 (89%)	
Pack-years (*n*)	30 (11–40)	30 (10–40)	30 (30–42)	0.311
COPD (*n*)	37/75 (50%)	33/66 (50%)	4/9 (44%)	1.000
LAAs category				
trace emphysema (LAAs ≤ 0.5%; *n*)	52/75 (69%)	48/66 (72%)	4/9 (44%)	0.122
mild emphysema (LAAs 0.5–5%; *n*)	14/75 (19%)	12/66 (18%)	2/9 (22%)	0.671
moderate emphysema (LAAs ≥ 5%; *n*)	9/75 (12%)	6/66 (10%)	3/9 (34%)	0.070
LAAs surgery lobe/LAAs whole lung (%)	10 (0.25–24)	11 (0–25)	9 (0.75–12)	0.445
Cancer pathological stage				
stage IA (*n*)	53/75 (71%)	47/66 (71%)	6/9 (67%)	0.716
stage IB (*n*)	10/75 (13%)	9/66 (14%)	1/9 (11%)	1.000
stage II (*n*)	5/75 (7%)	5/66 (8%)	0/9 (0%)	1.000
stage III (*n*)	7/75 (9%)	5/66 (7%)	2/9 (22%)	0.195
Cancer histology				
adenocarcinoma (*n*)	60/75 (80%)	53/66 (80%)	7/9 (78%)	1.000
squamous cell carcinoma (*n*)	9/75 (12%)	7/66 (10%)	2/9 (22%)	0.293
other (*n*)	6/75 (8%)	6/66 (10%)	0/9 (0%)	1.000
Upgrade in LAAs category between staging and follow-up CT scans (*n*)	19/75 (25%)	17/66 (26%)	2/9 (22%)	1.000
Difference in LAAs between staging and follow-up CT scans (%)	−0.01 (−1.74 ± 0.18)	−0.04 (−1.78 ± 0.10)	0.11 (−1.23 ± 15.12)	0.142

Data are shown as median and interquartile range in brackets for continuous variables while as absolute number and percentage in brackets for categorical variables. Abbreviations: COPD, Chronic Obstructive Pulmonary Disease; LAAs, low attenuation areas; NSCLC, nonsmall-cell lung cancer.

**Table 2 life-13-01377-t002:** Cox proportional-hazards regression analysis for patients’ survival.

Variable	Univariable Analysis		Multivariable Analysis	
	HR (95% CI)	*p*-Value	HR (95% CI)	*p*-Value
Age > 70 years-old	1.75 (0.47–6.52)	0.400	-	NR
Males	1.97 (0.41–9.43)	0.397	-	NR
History of other cancer	1.93 (0.52–7.20)	0.326	-	NR
Current or former smoker	1.53 (0.19–12.17)	0.687	-	NR
Pack years > 30	1.24 (0.31–5.01)	0.755	-	NR
COPD	0.85 (0.23–3.17)	0.816	-	NR
LAA category (reference trace emphysema LAAs ≤ 0.5%)				
mild emphysema (LAAs 0.5–5%)	1.58 (0.29–8.57)	0.595	-	NR
moderate emphysema (LAAs ≥ 5%)	5.27 (1.18–23.49)	0.029	7.27 (1.60–32.96)	0.010
LAA surgery lobe/LAA whole lung ≥10%	0.41 (0.10–0.96)	0.048	0.24 (0.05–0.94)	0.046
Cancer pathological stage III	4.35 (1.02–21.06)	0.041	6.50 (1.11–37.92)	0.038
Adenocarcinoma histology	1.51 (0.30–7.41)	0.610	-	NR
Upgrade in LAAs category between staging and follow-up CT scans (n)	1.34 (0.27–6.56)	0.712	-	NR
Difference in LAAs between staging and follow-up CT scans >−0.01%	1.54 (0.38–6.13)	0.541	-	NR

*p*-values highlighted in bold are statistically significant (*p* < 0.05); Abbreviations: CI, confidence interval; COPD, Chronic Obstructive Pulmonary Disease; CT, computed tomography; HR, hazard ratio; LAAs, low attenuation areas.

## Data Availability

The data presented in this study are available on request from the corresponding author. The data are not publicly available due to privacy policy restrictions.

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
