# Peer review of "Association between Mortality and Lung Low Attenuation Areas in NSCLC Treated by Surgery"

_life, 2023, doi:10.3390/life13061377_

Round 1

Reviewer 1 Report

Evidence has shown that the presence of low attenuation areas is a significant predictor of poor prognosis and increased mortality rates in NSCLC patients treated by surgery, however, the underlying mechanisms are not fully identified, most likely related to the extent of lung tissue damage, compromised lung function, and the presence of underlying lung diseases.

This study done by Davide Colombi, et al investigated the association between mortality and lung low attenuation areas (LAA) in NSCLC treated by surgery based on 75 patients, and found that patients with moderate emphysema (%LAA≥5%) at staging CT showed significant shorter OS. Although several studies have been done to show that low attenuation of lung parenchyma can predict survival in patients with NSCLC cancer, this study could make some contribution to this field. The manuscript is well prepared, and data presentation supports the conclusion. This work could be publishable. I would recommend making a minor revision.

Minor comments:

In Fig 3, I would suggest making some changes of X axis value into 12 months scale up to 72 months, as the longest survival is 70 months. The curves may be looking a bit better.  It would be good to include the numbers of patients in each group.

Fig 3b, “more or equal 5%”: should be “more than or equal to 5%”; Fig 3c, “less or equal than 10%”: less than or equal to 10%.

There seems to be a lack of a conclusion. In the abstract, it seems that the conclusion is describing the results. If you agree I would recommend adding a concluding sentence something like this, The LAA ratio to the whole lung at staging CT could be a critical factor to predict overall survival of the NSCLC patients treated by surgery.

Author Response

Dear Reviewer,

thank you for reviewing our manuscript and for your insightful suggestions.

Attached down below our point-by-point response.

  1. In Fig 3, I would suggest making some changes of X axis value into 12 months scale up to 72 months, as the longest survival is 70 months. The curves may be looking a bit better. It would be good to include the numbers of patients in each group.

Response: thank you for this remark. We have scaled down to 70 months the maximum value of the x-axis, considering the longest survival of 70 months, as suggested. Additionally we have included in figures the number of patients in both groups.

  1. Fig 3b, “more or equal 5%”: should be “more than or equal to 5%”; Fig 3c, “less or equal than 10%”: less than or equal to 10%.

Response: thank you for this remark. We have amended the legends as suggested.

  1. There seems to be a lack of a conclusion. In the abstract, it seems that the conclusion is describing the results. If you agree I would recommend adding a concluding sentence something like this, The LAA ratio to the whole lung at staging CT could be a critical factor to predict overall survival of the NSCLC patients treated by surgery.

Response: thank you for this important remark. We have amended the conclusions of both abstract and at the end of the text with the sentence suggested.

Reviewer 2 Report

I would like to thank the handling editor for giving me the opportunity to review the manuscript entitled “Association between mortality and lung low attenuation areas in NSCLC treated by surgery” by Colombi and colleagues, which is currently under consideration for publication in Life. I would also like to commend the authors for their scholarly work, which presents a single-centre, retrospective cohort study on the impact of low attenuation areas (LAAs), indicative of emphysema, on the overall survival (OS) of patients undergoing lung resection for non-small cell lung cancer (NSCLC). The study utilized computed tomography (CT) scans for quantifying LAAs, with patients categorized based on the percentage of LAAs relative to the whole lung volume (%LAAs). The categories included trace emphysema (%LAAs ≤0.5%), mild emphysema (%LAAs 0.5-5%), and moderate emphysema (%LAAs ≥5%). The study reveals a significant association between higher %LAAs and shorter OS. Specifically, patients with moderate emphysema (%LAAs≥5%) at staging CT demonstrated significantly shorter OS. However, the study also identified a trend towards longer OS for patients with a higher %LAAs lobe ratio (≥10%) at staging CT, suggesting that the distribution of emphysema in the lung lobes may also play a role in patient outcomes. The study further explores the correlation between LAAs at quantitative CT and mortality in resected NSCLC. It suggests that in patients suitable for surgery, a %LAAs≥ 5% relative to the whole lung at staging CT is associated with shorter overall survival, while a higher %LAAs lobe ratio is associated with longer OS.

The study under consideration situates itself within the existing literature by addressing a significant gap in the understanding of the prognostic role of emphysema in patients undergoing lung resection for NSCLC. While previous studies have explored the impact of chronic obstructive pulmonary disease and emphysema on the outcomes of lung cancer patients, this study offers a novel approach by focusing on the quantification of LAAs as a marker of emphysema and its correlation with OS. The novelty of the study lies in its methodological approach and the insights it provides. It employs CT scans to quantify LAAs, categorizing patients based on the percentage of LAAs relative to the whole lung volume. This approach allows for a more nuanced understanding of the impact of emphysema on patient outcomes, going beyond the binary presence or absence of emphysema. The study's findings have the potential to contribute to the field. By revealing a correlation between higher %LAAs and shorter OS, the study provides valuable prognostic information that can guide clinical decision-making. Furthermore, the identification of a trend towards longer OS for patients with a higher %LAAs lobe ratio at staging CT suggests that the distribution of emphysema in the lung lobes may also have prognostic significance. This insight opens new avenues for research and could potentially lead to more personalized treatment strategies for NSCLC patients with varying degrees and distributions of emphysema.

The study also acknowledges its limitations, including its retrospective nature and potential inaccuracies in lung volume and %LAAs measurements due to inspiratory levels. These limitations highlight the need for further research to validate and expand upon the study's findings. Despite these limitations, the study represents a significant step forward in the understanding of the prognostic role of emphysema in NSCLC patients undergoing lung resection, and its findings have the potential to inform clinical practice and future research in the field.

The manuscript presents a well-conducted study with significant findings. However, there are several areas where improvements could be made to enhance the clarity, impact, and overall quality of the work:

1.      The introduction could benefit from a more detailed explanation of the current state of knowledge in the field. This would provide a clearer context for the study and its significance. Additionally, the authors could consider providing a more explicit statement of the research question or hypothesis at the end of the introduction.

2.      The methodology section is generally well-written, but it could be improved by providing more detailed information concerning the patient selection process and the criteria used to categorize patients based on the percentage of LAAs. This would enhance the reproducibility of the study.

3.      The discussion and conclusion sections could benefit from a more thorough interpretation of the results. The authors should consider discussing the implications of their findings for clinical practice and future research in more depth. Additionally, the conclusion could be strengthened by a succinct summary of the key findings and their significance.

4.      The authors have acknowledged several limitations of their study. However, they could consider discussing these limitations in more detail and suggesting ways to address them in future research.

In conclusion, I would like to reiterate my appreciation to both the editor and the authors for the opportunity to review this interesting and informative manuscript. I believe that the suggested modifications, if addressed, will further enhance the quality and impact of the work. I look forward to seeing the revised version and wish the authors success in their ongoing research endeavours.

While the manuscript is generally well-written, there are areas where the language could be made clearer and more precise. The authors should consider revising the manuscript for clarity, coherence, and conciseness.

Author Response

Dear Reviewer,

thank you for reviewing our manuscript and for your insightful suggestions.

Attached down below our point-by-point response.

  1. The introduction could benefit from a more detailed explanation of the current state of knowledge in the field. This would provide a clearer context for the study and its significance. Additionally, the authors could consider providing a more explicit statement of the research question or hypothesis at the end of the introduction.

Response: thank you for this important remark. We have expanded the context of the study and the aim of the research in the introduction. Particularly, we have better explained the current knowledge about the emphysema meaning for patients who underwent surgery in terms of functional impairment after resection and of mortality [Page 1-2, lines 38-58: “In patients affected by advanced squamous cell carcinoma higher visual score emphysema is associated with 2-fold risk increase of death [5]. The improvement of lung function after resection of cancer in patients with emphysema has been widely demonstrated [6,7]. When the tumor is located in highly emphysematous areas, cancer resection combined with “lung volume reduction surgery” (LVRS) determines an increase of the postoperative lung function [6]. Furthermore, in patients with emphysema who underwent lobar resection for cancer without LVRS, postoperative pulmonary function is better than predicted previous surgery [7]. In particular, patients with more pure obstructive airway disease with low forced expiratory volume in 1 second (FEV1) and low FEV1/forced vital capacity (FVC) ratio, have higher likelihood to benefit in pulmonary function after lobectomy [8]. The presence of emphysema is a predictor of death for lung cancer [9,10]. Nevertheless, the emphysema quantification at CT was estimated visually with semi-quantitative methods or with the software aid with a binary approach (presence or ab-sence of emphysema) [9,10]. The impact of software-aided emphysema quantification at CT (QCT) on overall survival (OS) at staging CT of patients resected for NSCLC has not been fully investigated].

We also better described the aim of our study [Page 2, lines 65-70: “Thus, the aim of our study was to test the association with OS of LAAs quantified at staging CT in patients who underwent radical surgery for NSCLC. In particular we aimed to test the prognostic significance in terms of OS for both LAAs extent relative to whole lung volume (considering three group of patients on the basis of LAA extent: %LAAs ≤0.5%, %LAAs 0.5-5%, and %LAAs ≥5%) and the LAAs volume of the lobe to resect relative to whole lung LAAs volume.”]

  1. The methodology section is generally well-written, but it could be improved by providing more detailed information concerning the patient selection process and the criteria used to categorize patients based on the percentage of LAAs. This would enhance the reproducibility of the study.

Response: thank you for this insightful remark. We have better described patient selection process and the criteria used to categorize patients on the basis of LAAs. In particular, we have better described how patients with suspected lung cancer are investigated in our hospital and surgical indication [Page 2, lines 79-84: “The indication for radical surgical treatment was decided after a multidisciplinary board with the participation of pulmonologist, thoracic surgeon, oncologist, radiotherapist, pathologist, and radiologist. All patients with clinical or radiological suspicion of lung cancer were discussed at the multidisciplinary board after the clinical staging process, performed on the basis of the current guidelines of the European Society of Medical Oncology (ESMO)”].

Additionally, we have better described the patients categorization on the basis of LAAs thresholds [Page 3, lines 144-148: “Each patient was categorized on the basis of %LAA thresholds at staging CT (Figure 1) as provided by the statement of the Fleischner society: a. group 1 with trace emphysema when %LAAs was ≤0.5%; b. group 2 with mild emphysema when %LAAs was included within 0.5 and 5%; group 3 with moderate emphysema when %LAAs was ≥5%”].

  1. The discussion and conclusion sections could benefit from a more thorough interpretation of the results. The authors should consider discussing the implications of their findings for clinical practice and future research in more depth. Additionally, the conclusion could be strengthened by a succinct summary of the key findings and their significance.

Response: thank for this main remark. We have modified the discussion including the comparison with other study that assessed prognostic significance of emphysema in patients treated surgically for NSCLC [Page 10, lines 535-541: “In patients affected by lung cancer, emphysema is a well-known risk factor for shorter OS [9,10]. Zulueta et al. demonstrated higher risk of death for lung cancer in patients with marked emphysema, defined visually as more than one-half of lung parenchyma [9]. In pathological stage I NSCLC, the presence of %LAA>10%, identified patients with 2-fold increased risk of death [10]. These findings are similar to the results obtained in the present study, which demonstrated reduced OS for patients with resected NSCLC and %LAAs≥5%.”].

Furthermore we have added a statement regarding the implications of our findings in clinical practice and future research; particularly we have underlined the tailored approach for surgery on the basis of lobar LAAs quantification at staging CT for a precise assessment of patients prognosis [Page 10, lines 551-556: “Our findings contribute to go beyond the binary presence or absence of emphysema for planning surgery in patients with lung cancer. Providing quantitative extent at CT of LAAs in both whole lung and in the lobe that should be resected, a more tailored approach for surgery could be achieved, with more precise prognosis assessment and prediction of postoperative lung function, particularly for patients with borderline pre-operative pulmonary function tests.”].

We have finally added, as suggested by reviewer 1, a final statement including the key finding of our study [Page 11, lines 594-595: “The LAA ratio to the whole lung at staging CT could be a critical factor to predict overall survival of the NSCLC patients treated by surgery”]

  1. The authors have acknowledged several limitations of their study. However, they could consider discussing these limitations in more detail and suggesting ways to address them in future research.

Response: thank you for this important remark. We have discussed in more detail the limitations of the study; in particular the lacking of correction for TLV of the emphysema quantification, the use of different scanners and kernels for CT acquisition and reconstruction, and the absence of qualitative emphysema analysis. We have also suggested future research details to overcome the limitations of the present manuscript [Page 11, lines 565-589: “This study carries several limitations. First, it was a retrospective analysis from a single institution. Second, the measurements of lung volume and %LAAs can be affected by inspiratory levels, since adjustment for total lung volume (TLV) was not performed; this limitation could determine an overestimation of %LAAs, considering that functional lung can be not completely ventilated due to sub- optimal inspiration, with consequent reduced whole lung volume [28]. Nevertheless, patients received detailed instructions for optimal inspiration before examinations, and CT scans affected by inadequate inspiration were excluded [28]. Third, CT scans were performed with different scanners and kernels. LAAs quantification can be affected by several parameters among which iterative re-construction algorithm, software versions, and noise reduction algorithm [29]. Particularly the use of noise reduction algorithm reduce the quantification of %LAAs, with better correlation to pulmonary function tests [29]. For this reason we applied a noise reduction algorithm in order to optimize emphysema quantification, despite CT were acquired with different scanner and reconstructed with different kernels [29]. Fourth, the visual analysis of emphysema characteristics as proposed by Lynch et al. was not assessed [22]. In smokers visual scoring of thoracic CTs provides independent prognostic information for the clinical management of ever-smokers; particularly mild centrilobular emphysema, moderate centrilobular emphysema, and confluent emphysema were associated with shorter OS. Nonetheless, the focus of our study was to assess the association of LAAs at quantitative CT with mortality in resected NSCLC, and not with qualitative emphysema analysis [22]. Fifth, the correlation with pulmonary function tests was not investigated, however, lung function of all patients was preserved at staging and permitted lung re-section. Future prospective studies in multicentric scale, including qualitative emphy-sema analysis and correlation with both preoperative and postoperative pulmonary function tests are required to confirm and strengthen the present findings.”].

  1. While the manuscript is generally well-written, there are areas where the language could be made clearer and more precise. The authors should consider revising the manuscript for clarity, coherence, and conciseness.

Response: thank you for this remark. We have revised manuscript language as suggested.
